# Comparative Amino Acid Profile and Antioxidant Activity in Sixteen Plant Extracts from Transylvania, Romania

**DOI:** 10.3390/plants12112183

**Published:** 2023-05-31

**Authors:** Andreea Maria Iordache, Constantin Nechita, Paula Podea, Niculina Sonia Șuvar, Cornelia Mesaroṣ, Cezara Voica, Ramona Bleiziffer, Monica Culea

**Affiliations:** 1National Research and Development Institute for Cryogenics and Isotopic Technologies, 4 Uzinei Str., 240050 Râmnicu Vâlcea, Romania; andreea.iordache@icsi.ro; 2National Research and Development Institute for Forestry “Marin Dracea” Calea Bucovinei, 73 Bis, 725100 Campulung Moldovenesc, Romania; 3Chemistry Department, Faculty of Chemistry and Chemical Engineering, Babeș-Bolyai University, Arany Janos 11, 400028 Cluj-Napoca, Romania; 4National Institute for Research and Development in Mine Safety and Protection to Explosion, 32-34 General Vasile Milea Str., 332047 Petroșani, Romania; 5Department of Biophysics, Faculty of Pharmacy, George Emil Palade University of Medicine, Pharmacy, Science and Technology of Târgu Mureș, 38 Gh. Marinescu Str., 540139 Târgu Mureş, Romania; 6National Institute for Research and Development of Isotopic and Molecular Technologies, 67-103 Donat Str., 400293 Cluj-Napoca, Romania; 7Biomolecular Physics Department, Faculty of Physics, Babeș-Bolyai University, Kogălniceanu 1, 400084 Cluj-Napoca, Romania

**Keywords:** amino acids, antioxidant activity, plants, synergy, food antioxidant supplementation

## Abstract

In addition to the naturopathic medicines based on the antiseptic, anti-inflammatory, anticancer, or antioxidant properties of plant extracts that have been capitalized upon through the pharmaceutical industry, the increasing interest of the food industry in this area requires potent new materials capable of supporting this market. This study aimed to evaluate the in vitro amino acid contents and antioxidant activities of ethanolic extracts from sixteen plants. Our results show high accumulated amino acid contents, mainly of proline, glutamic, and aspartic acid. The most consistent values of essential amino acids were isolated from *T. officinale*, *U. dioica*, *C. majus*, *A. annua*, and *M. spicata*. The results of the 2,2-diphenyl-1-pycrylhydrazyl (DPPH) radical scavenging assay indicate that *R. officinalis* was the most potent antioxidant, followed by four other extracts (in decreasing order): *T. serpyllum*, *C. monogyna*, *S*. *officinalis*, and *M. koenigii*. The network and principal component analyses found four natural groupings between samples based on DPPH free radical scavenging activity content. Each plant extracts’ antioxidant action was discussed based on similar results found in the literature, and a lower capacity was observed for most species. An overall ranking of the analyzed plant species can be accomplished due to the range of experimental methods. The literature review revealed that these natural antioxidants represent the best side-effect-free alternatives to synthetic additives, especially in the food processing industry.

## 1. Introduction

The COVID-19 pandemic crisis has raised new challenges related to finding natural remedies capable of fortifying the human body with increased natural resistance to various diseases [1,2]. One unique framework developed using antioxidants can stabilize and deactivate free radicals before they can interact with cells. Natural antioxidants, either raw extracts or their chemical constituents, are also effective in preventing causes of stress. The derived drugs are safer than synthetic nanomaterials; thus, herbs are thoroughly evaluated for their toxicity profiles [3,4,5]. Furthermore, plant remedies have more impact than synthetic ones due to their natural relationship with the human body and the environment [6]. The antioxidant activities of plant extracts were evaluated using numerous in vitro and in vivo tests based on features such as neutralizing free radicals and inhibiting lipid peroxidation [7].

The relationship between daily dietary ingestion and human health has increased the interest in biologically active compounds found in different plant species [8,9,10]. Plants are extensively used in nutrition, phytoremediation, and cosmetic products, as well as in insecticides, food flavorings, essential oils, and, most importantly, medicine [11,12,13,14,15,16,17]. Aromatic plants have been known and intensively used throughout history in therapy as medicines to treat or prevent various afflictions (e.g., rheumatic disorders, renal lithiasis, urinary infections, digestive diseases, and cardio-respiratory diseases), even since the emergence of drugs [18,19,20]. Medicinal herbs and extracts of their active components show therapeutic benefits that include anticancer, hepato-protective, antiviral, antidiabetic, cardiovascular disease-preventative and anti-leukemic properties [21,22,23,24,25]. Moreover, plant extracts are extensively used for health maintenance, especially as adjuvants in various food products [17,21,26,27]. Generally, plant organs and different plant extracts are growth-promoters due to their specific active substances [28,29]. A large spectrum of modern drugs has been synthesized following comprehensive research on the mechanisms of action of chemicals in medicinal plants [30,31]. Consequently, medicinal herbs play a significant role in essential medicine, and they are widely used around the world in their traditional form [32].

The literature shows extensive work exploring the bioactive compounds in plants. *R. officinalis*, *Z. officinale*, *S. officinalis*, *C. monogyna*, *T. serpyllum*, *M. koenigii*, *P. coronarius*, *M. spicata*, *C. majus*, *S. nigra*, *A. annua*, *C. carvi*, *S. officinale*, *T. officinale*, *U. dioica*, and *O. basilicum* have been comprehensively investigated for their chemical compositions and pharmacological properties. All plants studied are perennial herbs used mainly for medicinal purposes and for flavoring in the food industry. Five plants represent the Lamiaceae family, which contains high amounts of hydroxycinnamic acids (HCAs). *S. officinalis*, *C. monogyna*, *T. serpyllum*, *M. spicata*, *C. majus*, *S. nigra*, *C. carvi*, *S. officinale*, *T. officinale*, and *U. dioica* are featured in the Romanian flora, and they are also intensively cultivated in farms. It is known that phytochemicals vary with environmental stress conditions as a result of the activation of genes responsible for their biosynthesis. The adaptation of plants to salty soil, environmental pollution, or other restrictive biotic and abiotic factors has previously been associated with higher amounts of antioxidants in plants [33,34,35,36,37,38]. Recent investigations regarding the influence of environmental stressors on *Rosmarinus officinalis*, *Mentha pulegium*, and *Thymus vulgaris*’ chemical compositions demonstrate an increasing percentage of secondary metabolites in the year following a drought [39]. The authors found higher contents of coumarin, saponins, tannins, alkaloids, and flavonoids [39]. In Romania, there is a national program encouraging the collection of secondary products from the forest, such as plants that are then further processed in the medical or food industry. A selection of non-indigenous plants was chosen, and their antioxidant properties and amino acid contents were evaluated for a comparison between imported products and those of local origin.

However, various combinations including multiple local species, which can be assessed as food supplements that can reinforce the human body in the fight against autoimmune disease, must be developed. Numerous applications for the plants mentioned above are known, but these plants have been less broadly investigated in Romania, resulting in a lack of information. Thus, the present study aims to fill this gap by evaluating sixteen indigenous and exogenous plant species. The biological material was collected from natural sites in Romania/Transylvania, and two samples were taken from local markets, to create chemical composition profiles and compare the total and essential amino acid contents. Here, we have analyzed antioxidant activity using our results and a systematic literature review, in order to outline the main compounds responsible for their activity.

## 2. Results and Discussion

### 2.1. Amino Acid Quantification and Interspecies Comparison

The free amino acids investigated are classified based on their effects on food flavors in three categories: (i) umami (aspartate, glutamate, 4-aminobutyric, ornithine), (ii) sweetness (serine, alanine, glycine, threonine), and (iii) bitterness (tyrosine, leucine, valine, methionine, isoleucine, phenylalanine, proline, lysine) [40,41,42]. The contents of the sixteen amino acids analyzed had different patterns in each sample (Figure 1). Statistically significant differences in the total amino acid content were noted only between Pro and other amino acids. The lowest levels were found for methionine, ornithine, tyrosine, leucine, and serine. The highest values (mg/g) among the essential amino acids were found for proline in *M. koenigii* (18.80), *T. officinale* (8.0), *A. annua* (7.71), *C. majus* (2.38), *S. officinalis* (1.68), and *R. officinalis* (0.32) (Figure 1A). Proline is a proteinogenic amino acid essential to the primary metabolism, and it accumulates in the cell cytoplasm as a response to environmental stress [43]. It is a low-temperature cryoprotectant that binds the intracellular water and protects the structure and functions of different enzymes and proteins by preventing ice formation inside the cell [44]. In addition, the proline was found to be essential to the scavenging of free radicals and reducing the cellular redox potential under stress conditions. Several applications in the cryoprotection of biological samples, such as human stem cells or plant tissues, based on this osmolyte action are known [45]. Kumari et al. indicated proline as the most prevalent amino acid in *M. koenigii* [46]. Salt injury and high K concentration, coupled with high Zn availability due to salt stress, are the main factors associated with high leaf proline content in *T. officinale* and *A. annua* [35,36]. Other stress factors, such as drought [37], heavy metal toxicity [33], oxidative stress [34], and abiotic stress [38], are often found to be associated with high Pro contents in plants.

The amino acid with the second highest abundance (mg/g) was Glu, found in *Mentha spicata* (4.81), *P. coronarius* (1.63), *T. serpyllum* (1.01), *S. officinale* (0.72), *C. carvi* (0.70), and *C. monogyna* (0.52) (Figure 1A). Glu regulates photosynthesis, the carbon/nitrogen cycle, Ca^2+^ homeostasis, root shape, defense signals, and drought or salt conditions [47]. Glutamate-like receptors (GLRs) can be activated through glutamate action, along with glycine and serine, and are inhibited or blocked by their antagonists (e.g., lanthanum ion or magnesium ion) [48]. Intensive studies over the past few decades have investigated plant physiology and biogeochemistry in relation to molecular genetics in seeking to understand the regulatory mechanism and nitrogen supply process in plant growth and development. The final inorganic nitrogen form available to plants (ammonium) is assimilated into an organic molecule (2-oxoglutarate) via a process that combines the action of glutamine and glutamate synthetase enzymes, which are further used as amino acid donors [49]. Glutamate has a dual role in plant metabolism as a glutamate receptor and a transmembrane protein associated with defense response. Recent studies have indicated that glutamate triggers GLR3-dependent Ca^2+,^ and when it is present at high external concentrations, calcium response is depleted in the leaves [50]. The over-accumulation of glutamate can be an endogenous indicator of tricarboxylic acid cycle suppression under excess soil ammonium conditions [51].

The amino acid with the lowest abundance (mg/g) was Asp, found in *U. dioica* (3.16), *Z. officinale* (2.30), *S. nigra* (1.73), and *O. basilicum* (0.06) (Figure 1A). Asp is a significant component of proteins and an active residue in various enzymes, acting as a precursor of biomolecule synthesis. Asp is responsible for plant development and adaptation to restrictive environmental factors. Further, aspartate is a precursor for methionine, threonine, lysine, and isoleucine, which are essential amino acids, and it is essential in glutamate-to-glutamine conversion [52]. Even arginine, tyrosine, and phenylalanine are involved in biosynthesis. Aspartate acts as (i) an endogenous metabolic limiter for cell proliferation, (ii) a coordinator of nitrogen assimilation into amino acids, (iii) a drought stress-specific responsive metabolite, and (iv) a biomarker of biotic and abiotic stress-inducing environmental exposure [53]. It has been found, along with asparagine and glutamine, to play an essential role in nitrogen fixation in germinated seeds [54]. Asp is usually associated with Pro, Glu, Ser, and Ala, as an indicator of alkaline salt stress [55].

Six essential amino acids (EAAs) that are indispensable for humans and cannot be synthesized from metabolic intermediates, including threonine, valine, leucine, isoleucine, methionine, phenylalanine, and lysine, were investigated. They showed the following contents (mg/g) in decreasing order: *T. officinale* (1.72), *U. dioica* (1.35), *C. majus* (1.27), *A. annua* (1.23), *M. spicata* (1.07), *S. nigra* (0.94), *Z. officinale* (0.87), *P. coronarius* (0.56 mg/g), and *O. basilicum* (0.05). In terms of nutrition, these amino acids can only be derived from an exogenous diet and are obtainable from a single complete protein found in plants, such as in soy. Thus, it is usually derived from animal-based sources containing all nine essential amino acids. A critical distinction between fresh and processed animal proteins is that, in the latter, it is derived from a mixture of artificial ingredients (sugar, sodium), which are associated with multiple diseases. On the other hand, the total free amino acid (TAA) contents were the highest in *M. koenigii* (20.70), *T. officinale* (13.47), and *A. anua* (12.08). This indicates that the plants investigated contain notable amounts of amino acids under natural conditions. Moderate contents were found in *M. spicata* (9.07), *U. dioica* (7.40), *C. majus* (7.03), and *Z. officinale* (5.78). In other plants, the contents varied from the level found in *S. nigra* (4.30) to that in *O. basilicum* (0.26). Following joint FAO/WHO/UNU recommendations, a ratio of 83.5 mg/(kg·D) EAA is suggested for adult consumption. We find that this endorsement is entirely appropriate, considering our findings of EAA/TAA ratio values between 0.22 mg/g (*S. nigra*) and 0.02 mg/g (*M. koenigii*) (Figure 1B).

### 2.2. Antioxidant Activity

Kinetically assessing the evolution of AA in plant extracts over time can be used to infer each extract’s behavior. The antioxidative system is associated with the lifetime of reactive oxygen species in the cellular environment [56]. Plants’ antioxidant properties are frequently associated with their polyphenolic contents of aromatic amines, phenolic acids, essential oils, flavonoids, proanthocyanins, and bioactive compounds. Our results highlight various phytochemical compounds associated with antioxidant activity in the sixteen plants investigated, enumerating terpenes, flavonoids, and phenols. All sixteen plants studied showed significant antioxidant activity, thus representing a cumulative source within a varied daily diet. The scavenging effects of each plant extract on the DPPH radical were calculated, and four patterns can be highlighted (Figure 2). The first group included *R. officinalis*, which achieved its maximum activity before 1 min; this was followed by *S. officinalis* and *Z. officinale*, which showed the most intense activity after 1 min (95.02, 81.21, and 76.77%) and after 5 min (86.79, 92.33%) (Figure 2A), respectively. Both *R. officinalis* and *S. officinalis* showed high contents of eucalyptol and rosmarinic acid, and *Z. officinale* contained terpenes, zingiberene, and phenolic compounds, such as gingerols, as primary compounds.

The second group, containing *C. monogyna*, *P. coronarius*, *T. serpyllum*, and *M. koenigii*, showed promising activity after 1 min (31.91, 47.02, 44.50, 43.38, and 46.04%, respectively), which progressively increased up to 30 min, after which the values 67.09, 92.95, 89.00, and 91.41% were found (Figure 2B). The defining characteristic of all species in this group is the presence of phenolic compounds that have a beneficial effect on human health, enacted through their antioxidant activity. The investigations show that *C. monogyna* was rich in flavonoids, such as quercetin and rutin, *P. coronarius* contained citral and limonene, *T. serpyllum* had high amounts of thymol, carvacrol, and rosmarinic acid, and the phenolic content of *M. koenigii* was associated with flavonoids and phenolic acids. The third group included *A. annua*, *M. spicata*, *S. nigra*, and *C. majus*, which all showed a good antioxidant capacity after 1 min (17.88, 27.52, 32.92, and 30.91%), while after 30 min this activity did not exceed 35.19, 55.81, 56.72, and 67.09% (Figure 2C). *A. annua* contained volatile constituents such as camphor and germacrene, and non-volatile ones including flavonoids and coumarins. *M. spicata* contained a variety of volatile compounds, including menthol, carvone, limonene, and polyphenols (rosmarinic acid). The compounds identified in *S. nigra* included various terpenes, such as limonene, α-pinene, β-pinene, myrcene, and linalool, as well as phenols and flavonoids. Finally, *C. carvi*, *O. basilicum*, *T. officinale*, *S. officinale*, and *U. dioica* showed low antioxidant activities, which after 30 min reached only 12.30, 8.04, 25.75, 19.67, and 10.50% (Figure 2D). Despite the fact that the antioxidant activities of plant extracts in this group were relatively low here, various studies have indicated their intense beneficial effects on human health in relation to different diseases. *C. majus*, *T. officinale*, *S. officinale*, and *U. dioica* are not aromatic plants, and the contents of volatile compounds therein were minimal. Even so, these plants contained alkaloids with a relatively low content of volatile compounds. *C. carvi* and *O. basilicum* were rich in volatile oils, but the main compounds, linalool, carvone, and limonene, demonstrated low antioxidant potential here.

To evaluate the natural groupings between samples based on DPPH free radical-scavenging activity, network analysis (Figure 3A) and principal component analysis (PCA) (Figure 3B) were selected to investigate the data structure. Network analysis is used for data visualization to obtain insights regarding the most influential nodes and detect gaps in the studied data. The basis of the relationship was the co-occurrence of similar DPPH free radical-scavenging activities analyzed in groups of intervals. If the values fit into the mean interval, they were assigned a more substantial weight, and decreased to the extremes of the interval. *Z. officinale* and *R. officinalis* were distinguished from the other plants based on their extreme antioxidant properties. Even so, the distinct multiplicities interacting with one another show heterogeneous patterns of DPPH values. Essentially, four specific groups tend to co-occur next to each other. Structural gaps are illustrated as empty spaces between the clusters of interconnected nodes and indicate areas for potential segregation between different plant species’ potential. The results correlate with those derived from the literature review, showing that several plants have been extensively studied and used in various industries. Some others, despite having indicated in various studies an antioxidant potential, are less widely investigated, and mainly assessed in vitro, with recommendations made for further detailed investigations.

PCA was applied to reduce dimensionality but preserve the maximum variability found in the dataset, and the results are illustrated in Figure 3B. The principal components explained 87.12% (PC1) and 8.62% (PC2) of the total variance. The results show similar patterns in the plant extracts demonstrating their antioxidant activity. Thus, the DPPH free radical scavenging can separate the extracts into two main groups—the first, with positive values, containing *R. officinalis*, *Z. officinale*, *S. officinalis*, *C. monogyna*, *T. serpyllum*, *M. koenigii*, and *P. coronarius*, the second including *M. spicata*, *C. majus*, *S. nigra*, *A. annua*, *C. carvi*, *S. officinale*, *T. officinale*, *U. dioica*, and *O. basilicum.* Each group was further separated into two subgroups and has been discussed following a literature review of the principal compounds biosynthesized for in vitro and in vivo utilization.

### 2.3. Compounds, Active Principles, and Industry Importance of the Local Plant Extracts—Literature Synthesis

Various pharmacological actions of *R. officinalis* have been assessed to determine the main differences between, and possible advantages of consuming, products that combine multiple plants to fortify the human body. The plant extracts are certificated as “generally safe” by the US Food and Drug Administration (FDA). The antioxidant activity of *R. officinalis*, used in essential oils, was previously reported to be derived partially from phenolic groups [57]. The antioxidant activities were not exclusively associated with phenols. This conclusion can be drawn based on the results showing a lack of correlation between rosmarinic acid (RA) and antioxidant activity [58]. Contrarily, most studies found a strong correlation between RA and antioxidant activity in *Salvia* species, demonstrating that the result can be further diversified depending on the plant species investigated [59]. Another investigation has shown that the anti-radical activity of *R. officinalis* leaf, measured in summer, correlated with total phenols, total flavonoids, condensed tannins, and carnosic acid [60]. Even so, a chemical analysis of the plant extract indicated the base compounds of RA, chlorogenic acid, and caffeic acid [61]. The full spectrum of compounds identified, including terpenes and phenolic compounds, are frequently associated with biological activities based on gallocatechol, betulinic acid, rosmadial, safficinolide, genkwanin, hesperidin, homoplantaginine, scutellarein, and carnosic acid [62]. Most of these compounds are associated with anti-inflammatory, antibacterial, neuroprotective, antitumoral, and anticancer actions [63,64,65].

The anticancer activity of *R. officinalis* plant extracts was demonstrated in various cancer cells, including colon (HT-29, HCT-15, CO-115, Ls174-T, CaCo2, SW-480, CT-26) [4,66,67], prostate (LNCaP, PC3, DU-145, 22RV1) [22], liver (Hep-3B, HepG2) [22], breast, leukemia, and pancreatic. The reviewed articles emphasize RA, followed by carnosic acid and carnosol, as essential groups with anticancer activities. RA was isolated from 39 plant families (especially taxa of Lamiaceae and Boraginaceae) and is known for its intense antioxidant activity [68]. RA absorption and metabolism in the human body resulted in only a tiny amount being degraded into caffeic acid, ferulic acid, and *m*-coumaric acid, which are eliminated in the urine [69]. Many studies have investigated the bioactivity of RA due to its anticancer, hepatoprotective, neuroprotective, antioxidant, anti-inflammatory, antiviral, antimicrobial, antimutagenic, and antiallergic actions [70]. The biological activities of RA are associated with antioxidant activity and membrane stabilization, with potential use in treating cancer, rheumatoid arthritis, and bronchial asthma, but not peptic ulcers [19]. RA has found wide applications in the food industry, based on its capacity to scavenge free radicals and chelate pro-oxidant ions [71]. The plant contains around 1% RA, depending on whether it is dry or wet, and considering the wide range of factors that determine plant RA content, from environmental factors to those related to plant growth and physiology. Based on the need for high-purity RA in pharmaceutical applications, the food industry, and cosmetics, multiple plant sources are required around the globe. *R. officinalis* plant extracts are commonly screened and are already being used as stabilizers in the food industry. Rosemary diterpenes (carnosic acid and carnosol) have been found to be resistant to pasteurization, thus preserving antioxidant capacity for the product’s entire shelf life [72]. The release rate of rosemary polyphenols in food indicates its limited efficiency in a lipophilic environment, but hydrophilic environments are beneficial in this regard [72,73]. Rosemary extract was significantly correlated with sunflower oil degradation and reduced the total polar compounds, free fatty acid, and acrylamide formation during deep-fat frying [8].

In the phenolic profile of *Z. officinale*, the five compounds most frequently present (caffeic acid, ferulic acid, RA, amentoflavone, and syringic acid) are significantly correlated with antioxidant activity [5,56]. Other compounds have been isolated, including isovanillin, beta-sitosterol palmitate, glycol monopalmitate, hexacosanoic acid 2,3-dihydroxypropyl ester, maleimide-5-oxyme, p-hydroxybenzaldehyde, adenine, 6-gingerol, 6-shogaol, 1-(omega-ferulyloxyceratyl) glycerols [74], and *O*-methyldehydrogingerol [75]. Some compounds, such as 6- and 8-gingerol, are suspected to be sensitive to temperature and capable of conversion into 6- and 8-shogaol under high-temperature stress [76]. The thermogenic properties of crude drugs derived from ginger are associated with the active compounds 10-gingerol, 10-shogaol, 10-gingerdiols, and 10-gingerdiols 3,5 diacetate [77]. Many meta-analyses and review articles have focused on clinical trials, as well as anti-inflammatory, analgesic, metabolic, antidiabetic, anti-ulcer, gastric, antisecretory, antispasmodic, larvicidal, immunomodulatory, and other beneficial actions [20,78,79]. The anticancer activity of *Z. officinale* was demonstrated in colon carcinoma HT-29 and HTC116 cells [80,81], cervical HeLa cells [25], breast MDA-Mb-231 and MCF-7 cells [25], lung A549 cells [25], and leukemia K-562 cells [25]. The nutritional compounds identified are proteins, carbohydrates, dietary fibers, fat, ash, iron, calcium, and carotene [9]. Monoterpenoids, sesquiterpenoids, aldehydes, and other unidentified compounds are involved in producing *Z. officinale*’s flavor and aroma [10]. Camphene, α-curcumene, α-farnesene, zingiberene, and sabinene are associated with its taste [82]. Since *Z. officinale* is known to be generally safe, it is present in the food industry in various forms, being found in spiced food, herbal drinks, and additives. Fresh rhizome consumption can be sustained with no severe side effects reported. Multiple studies have indicated a beneficial effect on stimulating bile acid production, the activity of digestive enzymes in the pancreatic lipase, small intestine mucosa, and reductions in bile salt secretion [83].

The chemical compositions of *C. monogyna*’s leaves, fruits, and flowers contain phenolic compounds, vitamins, and triterpenoids. Folk medicine recommends it for use for its hepatoprotective, cardioprotective, neuroprotective, anti-inflammatory, gastroprotective, and antimicrobial effects [32]. Clinical studies suggest hawthorn extract for use in conditions of heart failure, myocardial dysfunction, and cardiac insufficiency, based on its chemical profiles that relate to antioxidant activity [84]. The antioxidant compounds most frequently isolated are flavonoids, proanthocyanidins, sterols, and triterpene acids. Mass chromatography has shown that the hawthorn fruit extract contains flavan-3-ols monomers, dimers, trimers, tetramers, procyanidin B2, quinic acid, caffeic acid hexoside, flavonols, flavonol glycosides, anthocyanins, and epicatechin [85]. Vitamins, including tocopherols and ascorbic acid, specifically β-carotene, have been found in *C. monogyna* plant extracts. α-tocopherol is frequently cited for preventing various diseases and sequestrating free radicals. It has been noted that the leaves and flowers have higher antioxidant activities than fruits [86]. The eco-friendly and green synthesis of AgNPs nanoparticles from leaf extracts has been tested for its effects on various cancer cells (gastric; breast adenocarcinoma; non-small cell lung cancer; cervical carcinoma; hepatocellular carcinoma), showing a significant anticancer effect based on its phenolic contents [6,87,88]. One severe problem relates to the increasing pathogenic resistance to drugs, so eco-friendly approaches are required to improve the effectiveness of existing antimicrobial agents. Besides their antioxidant-scavenging effects, crude *C. monogyna* plant extracts showed an antifungal activity that effectively suppressed multidrug-resistant bacteria [89]. Their nutrient composition shows the prevalence of carbohydrates, proteins, fatty acids, fructose, glucose, sucrose, trehalose, minerals, calcium, vitamin C, carotene, and sugars [90]. Fruits are extensively used in the food industry as concentrated supplements and in cosmetics [11]. Extracts from hawthorn leaves, flowers, and berries can induce reduced oxymyoglobin oxidation in meat foods, indicating their promise in relation to the industrial production of functional ingredients with an antioxidant capacity [84]. Sausages with traditional compositions are popular in the commercial food industry [91]. Natural beverages derived from hawthorn fruits with high energy, vitamin, and mineral contents are pasteurized in Türkiye (pestil) and are consumed for their positive effects on human health [16]. Recent investigations have illustrated the physiological activities of polysaccharides obtained from *C. monogyna* as prebiotics and antioxidants with applicability in the food industry [17]. Baked hawthorn products, such as bread and jams, are recommended to consumers with type 2 diabetes. Flavoring agents obtained from hawthorn berries, containing enriched bioactive phenolic compounds, have demonstrated their effective utilization in beer and wine. The pectin extracted from *C. monogyna* in wine residue has shown applicability in yogurt and water kefir production.

*P. coronarius* has been insufficiently studied in relation to its antioxidant activity, despite being frequently used in traditional medicines for its hepatoprotective, antidiabetic, and inflammatory activities [92]. Its high flavonoid, triterpene, coumarin, gallic acid, and phenolic acid contents are responsible for its strong antibacterial effects [92]. The flavonoids quercetin, rutin, kaempferol, isorhamnetin, naringenin, and eriodictyol are the constituent compounds of the ethanolic extract derived from the leaves and flowers of this plant [92]. They determine its antioxidant capacity, and their activity is related to oxidative stress [93]. Other studies found the following phytochemicals in the plant’s leaves and flowers: luteolin 7-glucoside, 7-methoxy coumarin, chlorogenic acid, caffeic acid, delphinidin 3-rutinoside chloride, hyperoside, ferulic acid, RA, trans-p-coumaric acid, bergapten, myricetin, quercetin, and T-resveratrol [94]. Unfortunately, relatively few studies have discussed the antioxidant activity of these compounds, even if our results indicate high potential in this area.

*T. serpyllum* is rich in secondary metabolites (phenolic monoterpenoids carvacrol and thymol) and is frequently used for its expectorant, antitussive, antispasmodic, anthelmintic, and carminative properties [95]. It was found that the flavonoids (apigenin and luteolin derivates) and phenolic acids (cinnamic, carnosic, and RA) therein are associated with the antioxidant capacity of *T. serpyllum* [96]. Other studies on the methanolic extract of *T. serpyllum* isolated rare compounds, such as 3-ketopentatriacontanoic acid and 27-ketotriacontanol [97]. Terpenes were the most abundant volatile compounds isolated from wild thyme extracts [98]. Given its content of aromatically active compounds, namely thymol, sabinene hydrate, carvone, delta 3-carene, and myrcene, the plant has been extensively used in functional foods since ancient times, as main ingredients and as additives to improve flavor [98]. The thymol and carvacrol found in essential oil extracts have found specific commercial use as stabilizers, due to their high contents of antifungal and antibacterial compounds [99]. The methanol extract of *T. serpyllum* showed a strong antimicrobial activity [3]. Numerous studies have shown that the *T. serpyllum* plant extract is associated with significant cytotoxicity for cancer cells (MCF-7, MDA-MB-231, A549), while no effect has been observed on normal cells [3,100]. The essential oil extracted from thyme has a broad spectrum of pharmacological properties, including antirheumatic, anti-inflammatory, anti-allergic, expectorant, diuretic, and antiseptic [18]. The plant extract has shown the capacity to inhibit various proinflammatory mediators, including cytokines and enzymes [101]. Its essential oil inhibits the growth of *Penicillium* sp., *Aspergillus* sp., and *Botrytis* sp. [102], and in the vapor state, it can be used in the storage of root vegetables, emphasizing its effectiveness as a food preserver [103]. The levels of cytotoxicity produced by thymol and thyme essential oils are recognized as generally safe. Still, a recommended dose of 10 g of dried leaves per day is outlined, based on various in vivo and in vitro studies.

*M. koenigii* is a rich source of aromatic terpenes and carbazole alkaloids, and its antioxidant activity strongly correlates with its content of phenolic compounds [104]. The *M. koenigii* leaf extract demonstrated a stronger action in tests of microorganism activity compared to standard drugs, indicating its potential use in developing new antimicrobial, antioxidant, antidiabetic, and cytotoxic compounds [105]. The methanolic leaf extract of this plant caused a promising increase in the phagocytic index via the deletion of carbon particles from the blood system. Several phytochemical compounds were found in *M. koenigii*: 3-carene, α-thujene, camphene, allyl(methoxy)dimethylsilane, β-myrcene, α-terpinene, g-terpinene, mahanine, koenine, koenigine, girinimbiol, coumarin, murrayanol, murrayagetin, and marmesin-1′-O-rutinoside [106,107]. Its flavonoid compounds and phenolic acids are associated with its antioxidant activity [108]. The essential oil of *M. koenigii* showed an antimicrobial effect against *B. subtilis*, *S. aureus,* and *P. vulgaris* and an antifungal effect against *C. albicans*, *C. tropicalis*, *A. niger,* and *A. fumigates* [109]. In recent studies, the high abundance of phenolic compounds and flavonoids isolated from *M. koenigii*, administrated in the form of a fermented beverage, led to a rapid increase in protein and amino acid contents, which could be exploited in anticancer treatments [110]. In the food industry, *M. koenigii* leaf extracts have demonstrated their high potential for use as probiotics [27] and as bio-preservatives for fruit juice [21] or food additives [111]. During storage, curry leaf powder inhibits the development of oxidation products in meat patties [112]. The dried powder was also added to cookies to increase the contents of protein, dietary fiber, minerals, and β-carotene, as well as the radical scavenging activity [113].

The major compounds of *A. annua* are directly correlated with its habitat and include sesquiterpene lactones, flavonoids, coumarins, phenolic acid, tannins, saponins, fatty acids, proteins, oxygenated monoterpenes, artemia ketone, aliphatic compounds, essential oils, and camphor [28,114]. More than 600 bioactive compounds with high therapeutic potential have been found in extracts of this plant, with antioxidant, antitumor, antifungal, anti-inflammatory, antiprotozoal, and cytotoxic potential [115]. Artemisinin and plethora, both secondary metabolites, have shown a significant capacity to fight malaria [116]. Recently, newly developed artemisinin-derived compounds, e.g., dihydroartemisinin, have been tested for their use against malaria [115]. It was also demonstrated that the compounds in *A. annua* had potential inhibitory properties against coronavirus host proteins, related to artemisinin derivatives. It was shown that the compounds could reduce the SARS-CoV-2 burden and membrane fusion, as well as controlling host cells, and have a beneficial effect in modulating the host immune response to minimize infection damage [1]. The in vitro and in vivo anticancer activities indicate the high potential for using *A. annua* in non-small cell lung cancer treatment [7]. The extract showed viability for use against breast, pancreas, and prostate cancer, compared with normal epithelial cells, peripheral blood mononuclear cells, and lymphocytes resistant to treatment [117]. The dicaffeoylquinic acids derived from the plant’s leaf extracts inhibited the activities of the dipeptidyl peptidase IV inhibitor (DPPIVi), α-amylase, and α-glucosidase and have shown a promising effect on diabetes and its complications [118]. In the context of the food industry, dietary *A. annua* additives have been shown to improve the meat quality, antioxidant capacity, and energy status of stressed meat, via alterations in relevant mRNA expression [26].

In folk medicine, fresh and dry *M. spicata* leaves are used to treat gastrointestinal, respiratory, diuretic, and spasmodic issues. The pharmacological properties of these plant extracts and essential oils include antioxidant, antimicrobial, antidiabetic, and anticancer [119]. Their antioxidant activity is correlated with RA, luteolin glucoside, chlorogenic acid, rutin, apigenin, kaempferol, thymol, 1,8-cineole, eugenol, caffeic acid, quercetagetin 3,6-dimethyl ether, furmanic acid, luteolin, and tocopherol [120,121]. Carvone, limonene, and menthol act as analgesics with short-term effects. The phenolic and volatile compounds therein were found to have strong antimicrobial, antiviral, antioxidant, and antitumor effects [122]. Unfortunately, the findings from human clinical trials are limited, despite the fact that a positive effect on gastrointestinal tissue and the nervous system has been noted. The immunomodulatory and chemopreventive activities of these essential oils indicate their high plant potential for use in pharmacology, with restrictive effects on gastrointestinal reflux, kidney deficiency, and hiatal hernia [123]. The leaves are usually used in teas, beverages, syrups, flavorings, and ingredients (such as for lamb dishes). Menthol is added to various cosmetics and perfumes for its color and distinct odor. The tobacco industry uses *M. spicata* extracts to reduce throat discomfort and add a bitter taste. Its insect-repellant action and bio-pesticide applications make this plant suitable for use in other industrial sectors [12].

*S. nigra* represents a significant source of free amino acids, protein unsaturated fatty acids, vitamins, antioxidants, and minerals [124]. Fruits have shown biological activities based on polyphenols, anthocyanins, flavonols, phenolic acids, anthocyanins, and proanthocyanidins, which together promote the plant for medicinal use given its potential antioxidant activity [124]. The bacteria *M. luteus*, *P. mirabilis*, *P. fragii*, and *E. coli* are sensitive to *S. nigra* plant extracts, the most predominant phenolic acids therein being chlorogenic acid, sinapic acid, t-cinnamic acid, ferulic acid, coumaric acid, vanillic acid, and quinic acid, and the dominant polyphenols and flavonoids being rutin, catechin, kaempferol, and quercetin [125]. This plant’s anti-inflammatory and antioxidant activities have been demonstrated against cytotoxic *t*-BOOH concentrations and HepG2 cells, indicating their positive effects in preventing imbalances associated with excess reactive oxygen species and nitric oxide production [126]. The polyphenols and lectins derived from fruit extracts can inhibit SARS-CoV-2 and have demonstrated antidepressant and immune-boosting properties [127]. Colon cancer cells (RKO), breast tumor cell lines (MCF-7, MDA MB MB-231, Caco-2), lung adenocarcinoma cells (A549), and ovarian cancer cells (A2780) were subjected to cellular apoptosis after the application of an *S. nigra* plant extract [23,24]. Contrasting results were yielded when analyzing similar extracts and cancer cell lines, especially in relation to lung cell lines [24]. This confirms the need to further study the action of *S. nigra* plant extracts on various cancer cell lines. The food industry derives freshly pressed juices from fruits, with high capacities to reduce nitric oxide free radicals and enzyme over-activity [128]. Wine produced from elderberries has demonstrated intense antioxidant activity, as well as β-glucosidase and tyrosinase inhibition based on the compounds p-Hydroxybenzoic, protocatechuic acid, chlorogenic acid, quercetin, hexoside, and rutin [129]. In the wine-producing process, polycyclic aromatic hydrocarbons (PAHs) were most effectively inhibited by the polyphenols, which act by reducing the levels of free radical precursors. The addition of elderberry vinegar before grilling reduces exposure in humans to PAH contamination. Elderberry puree increases yogurt and kefir’s total solid, fiber, and carbohydrate contents.

*C. majus* is also known for its therapeutic potential, which is based on isoquinoline alkaloids. Several compounds have been isolated from the plant’s organs, including flavonoids, saponins, vitamins, mineral elements, alkaloids, acids, and their derivates. Unique to this plant is its high concentration of minerals and essential elements, especially potassium and phosphorus. Various pharmacological activities are associated with *C. majus* plant extracts, including antibacterial, antifungal, antiviral, antiprotozoal, anti-ulcer, hepatoprotective, radioprotective, antioxidant, anti-inflammatory, analgesic, anti-Alzheimer, and immunomodulatory. Even if its antioxidant activity has not shown extreme values, usually around 28–57% (ABTS and DPPH assay) [130], applicability in phytomedicine has been cited in the literature. Very high antimicrobial activities of the ethanolic extracts derived from *C. majus* against multidrug-resistant *Helicobacter pylori*, *Enterococcus faecalis*, and *Escherichia coli* strains have been determined [131,132]. The 8-hydroxydihydrosanguinarine and 8-hydroxydihydrochelerythrine alkaloids extracted from *C. majus* show a strong antifungal activity against clinical drug-resistant yeasts [133]. Recent studies have indicated that latex proteins could have pharmacological significance. Latex compounds were found to have a positive effect on human papillomavirus (HPV) infections and inhibited the expression of cancer cells (E6, E7) on the mRNA and protein levels [2]. Sanguinarine and berberine are isoquinoline and quinoline intercalator alkaloids that are recognized as strong antiviral agents, and they can be used to screen DNA fragments to inhibit the transcription and replication of other viruses, such as SARS-CoV-2 and human immunodeficiency virus [2]. Most studies have cited chelerythrine and chelidonine as essential in establishing strong antifungal activity against *Pseudomonas aeruginosa*, *Staphylococcus aureus*, and *Candida albicans*. Novel photosensitive agents, such as protoberberine, isolated from *C. majus* plant extracts, have been proven efficient in the apoptosis of HeLa, A5949, H460, HCT 116, SW480, MDA-MB 231, MCF-7, and C33A cancer cells [134,135]. Recent studies have raised questions regarding the risk-to-benefit ratio of using products containing *C. majus* plant extracts, which have been proven highly hepatotoxic due to the idiosyncratic metabolic reaction they induce.

*T. officinale* has been intensively studied, and the positive effects of extracts of this plant in the treatment and prevention of kidney immune system, viral, diabetic, bacterial, oxidant, neuro-compromised, pancreas, and cancer diseases were noted [136]. The plant is a valuable source of minerals, vitamins, and nutrients, even though other studies have found it difficult to evaluate its biological properties based on its compounds [137]. The homologous peptides ToHyp1 and ToHyp2 found in *T. officinale* flowers consist of eighteen proline residues with antifungal activity and which inhibit Gram-positive and -negative bacterial growth [138]. The anti-angiogenic, anti-inflammatory, and antinociceptive activities of *T. officinale* are induced by the saponins, flavonoids, alkaloids, phenols, methanol, and steroids that are highly concentrated in its stem, roots, and flowers [139]. The most commonly cited phenols with antioxidant activity are found in chicoric acid, chlorogenic acid, and caffeic acid, which are associated with oxidative stress reduction and the modulation of various blood processes. Only a few studies have discussed the potential of *T. officinale* for use in cancer therapies, such as for ovarian and liver cancer cells [140,141]. The cytokines and their corresponding receptors are responsible for the apoptosis of cancer cells. The consumption of plants is recommended for their antimutagenic and chemopreventive effects against environmentally induced carcinogenesis. In the food industry, products containing *T. officinale*’s petals, leaves, and roots, such as salads, marmalades, teas, drinks, coffee substitutes, and juice, are recommended for their beneficial health properties and nutritional value. However, toxicology and safety considerations must be made, given the gastric hyperacidity and contact dermatitis associated with repeated contact with plant-derived latex compounds.

*C. carvi* plant extracts contain several components known for their capacity to treat anxiety, physiological disorders, and depression, including carvacrol, carvone, *p*-cymene, gallic acid, quercetin, caffeic acid, syringic acid, neochlorogenic acid, α-pinene, limonene, γ-terpinene, quercetin-3-glucuronides, iso-quercitrin, kaempferol-3-glucoside, and polyacetylenic compounds [29]. The antioxidant activity evaluated through the DPPH assay was determined to be 8.37% [142], comparable with our results. The antioxidant activity found in several other studies was also high, mainly when a water–ethanolic extract was used, compared with a water extract. This further implies that plant extracts with high antioxidant activity (DPPH) can be obtained with an organic solvent. The antimicrobial activity of this plant was demonstrated against bacteria and fungi. The same study indicated sustained anticancer activity, with action against colorectal (Caco-2) and liver (HepG-2) cell lines [143]. A recent investigation demonstrated that a nanoemulsion of *C. carvi* essential oil could inhibit the proliferation of breast cancer cells (TUBO) and free radicals [144]. This seed extract oil was effective in the treatment of ulcerative colitis infections. Edible eco-friendly packaging films made of plant extracts are used for preserving food products and are composed of nanoparticles derived from *C. carvi* with antimicrobial and antioxidant properties [145]. *C. carvi*’s fruits, roots, and shoots are used in food and beverages. Oil extracted from its seeds is used in flavoring sweets given its high monosaturated and polyunsaturated fatty acid and oleic acid contents [146].

*S. officinale*’s antioxidant activity is correlated with sage polyphenols, flavone glycosides, carnosol, carvacrol, carnosic acid, caffeic acid, chlorogenic acid, *p*-coumaric acid, and RA derivates [147]. Methanolic extracts from various regions worldwide have shown potent antioxidant activity, with applications in medicine and the food industry. Their very high antioxidant activities and positive effects on human skin fibroblasts have been documented. The root extracts are recommended for the external treatment of joint distortions and myalgia. Allantoin is associated with triggering cell division, and comfrey is associated with treating hepatic cancer cells. Toxicology studies have revealed liver fibrosis associated with the consumption of this plant in humans, which is why it is used restrictively and only for a limited period. Its promising potential applicability as a food preservative is based on its high antimicrobial and antifungal activity, which can be achieved by carefully evaluating the constituent chemical compounds and their toxicology in humans.

*U. dioica*’s phytochemical composition comprises flavonoids, tannins, coumarins, lignans, sterols, and fatty acids [148]. In addition, phenols and polyphenols have been identified in its roots, stalk, and leaves, with high concentrations of naringin, ellagic acid, myricetin, and rutin. These have been cited as primary metabolites with antioxidant activity and pharmaceutical effects, including anti-inflammatory, antimicrobial, antidiabetic, cardiovascular, anti-ulcer, and antimutagenic [149]. Extracts of the plant are rich in nutritional components, including amino acids, fibers, phenolic compounds, vitamins, chlorophylls, carotenoids, fats, and minerals. The high variability in chemical composition between individual plants and *Urtica* sp. explains the differences in potential antioxidant response [150]. *U. dioica* is broadly used as a curative and as a functional food, based on its capacity to increase the antioxidant gene catalase, superoxide dismutase 1 and glutathione, and to suppress lipid peroxidation. In vivo studies have demonstrated the efficacy of using extracts from dried *U. dioica* leaves to inhibit oxidative stress in the brain. After application of the ethyl acetate fraction of the plant extract, a reduction in free electron accumulation and malondialdehyde (one of the final outputs of polyunsaturated fatty acid peroxidation in the cells) was noted in the striatum and cortex [151]. Positive effects of *U. dioica* plant extracts have been noted on liver regeneration via increased superoxide dismutase and the prevention of tissue alterations by the blocking of potentially harmful oxygen molecules within cells. Plasma lipoproteins and tissues’ susceptibility to glutathione depletion and lipid peroxidation was affected by the antioxidant capacity of *U. dioica* plant extracts, indicating their efficacy in treating blood disorders, including oxidative stress and blood pressure problems. Other studies analyzing malondialdehyde levels in the kidneys, muscle tissues, heart, liver, forestomach, and lungs demonstrated its positive effects on different enzymes, including those involved in antioxidant activity. *U. dioica* shoots, used in the form of herbs or spinach, can protect against obesity through lipid accumulation and glucose metabolism. Comprehensive reviews have analyzed the plant extract’s efficacy in treating autoimmune disorders (rheumatoid arthritis, allergy, eczema, and cancer) and obesity-induced insulin resistance.

## 3. Materials and Methods

### 3.1. Plant Material

Fresh biological materials were derived from sixteen wild plants in Transylvania, western Romania. The plant materials were identified at the National Research and Development Institute for Forestry (INCDS Marin Dracea, Câmpulung Moldovenesc, România). Materials from five samples were combined and mixed to obtain a representative sample for each plant. The investigated plants are detailed in Table 1.

### 3.2. Apparatus and Reagents—Sample Preparation

For the measuring of amino acids, a GC–MS technique was utilized. A Trace GC (DSQ Thermo Finnigan, San Jose, CA, USA) coupled with a quadrupole mass spectrometer was used to identify and quantify the amino acids. Separation was achieved using a non-polar Rtx-5MS capillary column with 5% phenyl methylpolysiloxane, with the parameters of 30 m length, 0.25 mm diameter, and 0.25 µm film thickness, and a temperature gradient program of 70 °C for 2 min, 5 °C/min increase to 110 °C, 10 °C/min increase to 290 °C, and 16 °C/min increase to 300 °C. Helium was used as the carrier gas at a flow rate of 1 mL/min. In total, 1 µL of each sample was injected in a split mode utilizing a TriPlus autosampler. The MS instrument was operated in electron ionization (EI) mode with an emission current of 100 µA and a mass range of 50–500 a.m.u at 70 eV; this was employed to identify the samples. The transfer line, injector, and ion source were all maintained at 250 °C. The plant extracts’ antioxidant capacities were assessed using a UV-Vis spectrophotometer, specifically the Agilent 8453 model.

### 3.3. Extraction Procedures

Amino acid extraction was performed using 100 mg of crushed dried plant parts. An ultrasound extraction using 1 mL of 6% trichloroacetic acid was performed. The extract was centrifuged at 6000 rpm for 5 min and purified on a Dowex 50W-W8 ion exchange column (Dowex^®^, Merck Millipore, Burlington, MA, USA). For the quantitative analysis, 50 µg of [15N]-glycine was used as an internal standard. The elution of amino acids was made using 4 M ammonium hydroxide. The carboxylic acid group was esterified for derivatization by treating the dried sample with 200 µL of butanol:acetyl chloride (4:1 *v*/*v*) for 1 h at 110 °C. This procedure was followed by the acetylation of the amine group using 100 µL of trifluoroacetic anhydride for 20 min at 80 °C.

### 3.4. DPPH Assay

In total, 100 mg of the crushed plant was extracted via ultrasound with 1 mL ethanol at 60 °C for 15 min. Further, the extracts were centrifuged at 7000 rpm and analyzed for antioxidant capacity. The free radical 2,2-diphenyl-1-picrylhydrazyl (DPPH) method was used to determine antioxidant activity. For the analysis, 10 μL (1 mg/mL plant) of each extract was used to decolorize a solution of 40 μM DPPH. The measurement was performed at 517 nm and was registered for 30 min using a spectrophotometer in the kinetic mode.

### 3.5. Analytical Performance of the Method

Amino acid standards were used for the validation of the method. This involved injecting standard solutions of amino acids, following the above-presented extraction and derivatization procedure. Each sample was produced following the same process. Suitable values for the validation parameters of linearity, precision, accuracy, and detection limit were obtained. The quantitative method manifested a statistically significant linear regression curve, produced using amino acid standards with known concentrations (0–100 µg/mL) and 50 µg/mL of an internal standard. The internal standard used was 15N-glycine (99 atom% 15N), and both the natural glycine and isotopomer required deconvolution and correction of the matrix calculation. Experimental measurements were undertaken to determine the fractional isotopic abundances of glycine and its corresponding labeled glycine ion (*m*/*z* 154 [M] and *m*/*z* 155 [M + 1], respectively). Matrix and regression curve calculations were employed to quantify the glycine. Precision and accuracy assessments were conducted for glycine using standards of 20 and 30 µg/mL (*n* = 7), yielding values lower than 6% and 11%, respectively.

The amino acids were eluted in the following order: alanine (Ala), glycine (Gly), threonine (Thr), serine (Ser), valine (Val), leucine (Leu), isoleucine (Ile), proline (Pro), methionine (Met), aspartate (Asp), phenylalanine (Phe), lysine (Lys), glutamate (Glu), 4-aminobutyric acid (GABA), ornithine (Orn), and tyrosine (Tyr). The standards underwent the above-described extraction and derivatization procedure (*n* = 3). The results of precision show a relative standard deviation (R.S.D.) below 20%. The linearity analysis of the amino acids yielded promising results, with a regression coefficient above 0.99 for almost all compounds. The accuracy was lower than 20%. The detection limit (LOD) was determined to be lower than 0.1 μg/mL.

### 3.6. Statistical Analysis

The experiments were performed in triplicate, and the data were statistically analyzed using IBM SPSS v27.0. Descriptive statistics for the continuous variables representing the properties of the chemical compounds have been presented as mean, standard deviation, minimum, and maximum. In addition, one-way ANOVA analyses and comparisons of means and variance were performed using Tukey and Levene’s tests for *p* < 0.05, in order to identify differences between the chemical profiles of different plant extracts. The amino acid content and antioxidant activity were subjected to principal component analysis (PCA).

## 4. Conclusions

This comparative investigation of plants’ chemical profiles represents a valuable case study regarding the potential of a range of plant species extracts to be used in the food industry, as well as in pharmacology and folk medicine. The present study highlights several properties of the sixteen plants investigated, which had significant amino acid contents and higher antioxidant capacities. Case reports from the literature have also been used to verify the optimal mode of utilization of each plant in the treatment of various human ailments and diseases, including antimicrobial, antibacterial, and antitumoral, as was emphasized by in vitro investigations. Here, we demonstrated that plant extracts from *M. koenigii*, *T. officinale*, *A. annua*, *M. spicata*, *U. dioica*, and *C. majus* have significant total free amino acid contents over 7 mg/g. *T. officinale* had the highest contents of essential amino acids, followed by *U. dioica*, *C. majus*, *A. annua*, *M. spicata*, *S. nigra*, *Z. officinale*, and *P. coronarius*. Proline was the most abundant amino acid, followed by glutamic acid, aspartic acid, lysine, glycine, and alanine. Plant extracts of *R. officinalis*, followed by *T. serpyllum*, *C. monogyna*, *S*. *officinalis*, and *M. koenigii*, had the highest antioxidant activity. Further toxicological studies are needed to evaluate the potential risk of oral consumption, mainly in relation to *S. officinalis*.

In our literature review, we addressed the chemical profiles of each of the species collected in Romania, which were then analyzed and compared with similar reports from around the globe. Despite our study indicating low antioxidant activity, the literature suggests that the presence of significant differences can be attributed to differences in the antioxidant assay synthesis method. Individual differences may be associated with restrictive local and regional environmental factors, as well as internal plant factors. The literature demonstrates that extracts of *R. officinalis*, *Z. officinale*, *P. coronarius*, *T. serpyllum*, *M. spicata*, and *S. officinale* have synergic effects based on the principal chemical compound that can be isolated, which is rosmarinic acid. Most plants evaluated showed a high phenolic content and substantial amounts of flavonoids, flavonols, and anthocyanins. However, establishing an overall ranking of the antioxidant capacities of the analyzed species is impossible, due to the different experimental methods used. Further, some plants (*P. coronarius*) still need to be further investigated, despite being widely used in folk medicine for preventive purposes, and they may also have a very high potential for use in the food industry.

## Figures and Tables

**Figure 1 plants-12-02183-f001:**
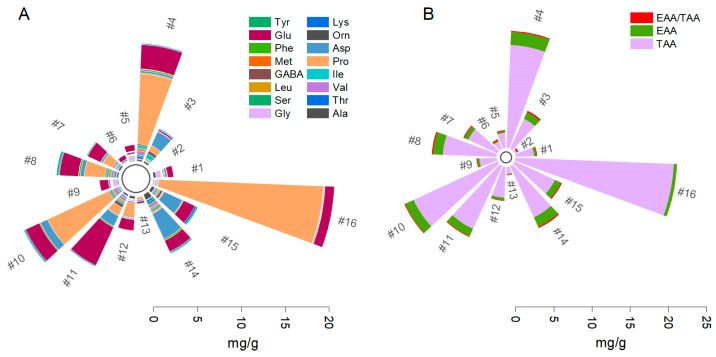
The amino acid contents in the sixteen plant extracts analyzed (**A**); essential amino acids (EAA), total amino acids (TAA), and the ratio of EAA/TAA (**B**). The numbers in the figure are associated with plant extracts (see Table 1).

**Figure 2 plants-12-02183-f002:**
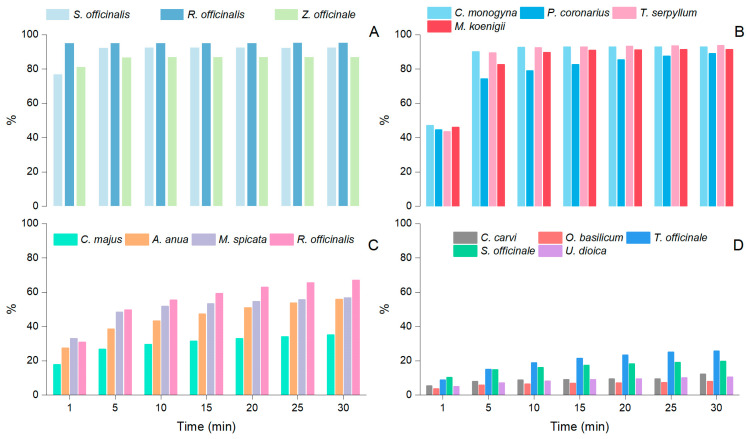
The antioxidant activities of the sixteen plant extracts investigated are presented. (**A**) The first group, including *S. officinalis*, *R. officinalis*, and *Z. officinale*; (**B**) the second group, including *C. monogyna*, *P. coronarius*, *T. serpyllum*, and *M. koenigii*; (**C**) the third group, including *A. annua*, *M. spicata*, *S. nigra*, and *C. majus*; and (**D**) the fourth group, including *C. carvi*, *O. basilicum*, *T. officinale*, *S. officinale*, and *U. dioica*.

**Figure 3 plants-12-02183-f003:**
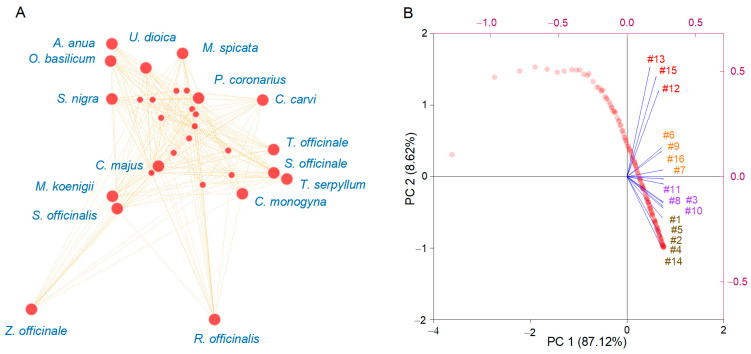
Network analysis (**A**) and principal component analysis grouping of sixteen plant extracts based on DPPH free radical scavenging (**B**); the numbers are associated with plants extracts (see Table 1).

**Table 1 plants-12-02183-t001:** The table presents the plant extracts investigated, the antioxidant activity (AOA) at 30 min, and the total amino acid (TAA) contents. The sample ID can be used to read the PCA analysis results.

Sample ID	Plants	Family	Parts Investigated	TAA	AOA at 30 min
#1	*Carum carvi* L.	Apiaceae	seeds	2.42	12.30
#2	*Ocimum basilicum* L.	Lamiaceae	leaf	0.26	8.04
#3	*Sambucus nigra* L.	Adoxaceae	fruits	4.31	67.09
#4	*Taraxacum officinale* (L.) Weber *ex* F.H. Wigg.	Asteraceae	leaf	13.50	25.75
#5	*Symphytum officinale* L.	Boraginaceae	root	1.94	19.67
#6	*Crataegus monogyna* Jacq.	Rosaceae	fruits	1.04	92.95
#7	*Philadelphus coronarius* L.	Hydrangeaceae	aerial plant	4.22	89.00
#8	*Chelidonium majus* L.	Papaverales	aerial plant	7.04	35.19
#9	*Thymus serpyllum* L.	Lamiaceae	aerial plant	2.26	93.76
#10	*Artemisia annua* L.	Asteraceae	aerial plant	12.09	55.81
#11	*Mentha spicata* L.	Lamiaceae	aerial plant	9.08	56.72
#12	*Salvia officinalis* L.	Lamiaceae	aerial plant	4.05	92.36
#13	*Rosmarinus officinalis* L.	Lamiaceae	aerial plant	0.95	95.16
#14	*Urtica dioica* L.	Urticaceae	aerial plant	7.41	10.50
#15	*Zingiber officinale* Roscoe	Zingiberaceae	root	5.82	86.86
#16	*Murraya koenigii* L.	Rutaceae	leaf	20.71	91.41

## Data Availability

All data relevant to the study are included in the article.

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
