# Peer review of "Comparative Amino Acid Profile and Antioxidant Activity in Sixteen Plant Extracts from Transylvania, Romania"

_plants, 2023, doi:10.3390/plants12112183_

Round 1
Reviewer 1 Report
Dear Authors,
your work entrusted to me for review is a very valuable scientific material. I find it very interesting from the point of view of science and usefulness to people. A small drawback of the work is the introduction. I think it's a bit chaotic. After line 55, I think that it would be good to justify the choice of plant species for research, where it came from and why these species were selected. It would be good to characterize them in terms of their availability in the environment, and at the same time indicate whether they can be widely exploited as herbal raw material.
Author Response
Dear reviewer,
We address sincere gratitude for a careful and thorough reading of the manuscript and the valuable comments and constructive suggestions that helped improve the draft's quality. We have revised our manuscript according to the recommendations; its final version is enclosed.
A small drawback of the work is the introduction. I think it's a bit chaotic. After line 55, I think that it would be good to justify the choice of plant species for research, where it came from and why these species were selected. It would be good to characterize them in terms of their availability in the environment, and at the same time indicate whether they can be widely exploited as herbal raw material.
Response: Thank you for your recommendation. The introduction chapter was rewritten. Also, several changes were performed in the Material and Method chapter, Results and Discussions.
Reviewer 2 Report
This is an interesting manuscript; however, it has two major concerns which are the identification of amino acids using GCMS. Amino acids are polar metabolites and are usually analyzed using LCMS. This explains the lower accuracy and detection limit (LOD). The second major concern is the authors explored the antioxidant activities of the samples (all plants have antioxidant properties), however, they have not annotated the chemical composition of the tested extracts.
Other minor concerns include:
- Plant names, tested parts along with the families should be assigned in a table instead of a text (lines 111-119)
- Section 3.3. Compounds, active principles, and industry importance of the local plants extracts literature synthesis contains a lot of redundant info; yes, the literature highlights those compounds in the tested extracts, however, the effect of the geographical origin, ecotypes, and seasonal variation could genuinely affect the chemical composition of the tested samples. This mandates that the authors should annotate their phytoconstituents.
Author Response
Dear Reviewer,
We address sincere gratitude for a careful and thorough reading of the manuscript and the valuable comments and constructive suggestions that helped improve the draft's quality. We have revised our manuscript according to the recommendations; its final version is enclosed. Point-by-point responses to the comments are listed below.
This is an interesting manuscript; however, it has two major concerns which are the identification of amino acids using GCMS. Amino acids are polar metabolites and are usually analyzed using LCMS. This explains the lower accuracy and detection limit (LOD).
Response: The amino acids are polar metabolites and can be analyzed using LC-MS, providing greater sensitivity and selectivity for such compounds. For GC-MS analysis of amino acids, a derivatization step is needed to increase the amino acid's volatility and hydrophobicity. Derivatization of amino acids is an essential step in analyzing amino acids by GC-MS. It improves the chromatographic separation of amino acids and enhances their detection sensitivity. There are also some other potential advantages to using GC-MS for amino acid analysis. One advantage of GC-MS is that it can provide very high-resolution separation of analytes, making it easier to identify and quantify amino acids, even in complex matrices.
Another advantage of GC-MS for amino acid analysis is that it can be relatively fast and efficient. GC-MS typically has a shorter run time than LC-MS, allowing for greater throughput and faster analysis of large samples. GC-MS can be a valuable tool for amino acid analysis in certain situations where LC-MS may not be feasible or desirable. Additionally, GC-MS can be less expensive and more readily available than LC-MS in some settings, making it a more practical choice for specific applications.
The isotopic labeled internal standard amino acid used for quantitative determination avoids losses due to extraction and derivatization and increases precision and accuracy. GC-MS can also be used for isotopic quantification, which measures a sample's relative abundance of isotopes. Using the isotope dilution technique, which involves adding a known amount of an isotopically labeled compound to the sample before extraction, the ratio of the isotopically labeled compound to the analyte is measured, and this ratio can be used to calculate the concentration of the analyte. For LC-MS analysis, some standard calibration curves are needed.
The second major concern is the authors explored the antioxidant activities of the samples (all plants have antioxidant properties), however, they have not annotated the chemical composition of the tested extracts.
Response: Thank you for raising this problem. We added this information to the text.
Other minor concerns include:
- Plant names, tested parts along with the families should be assigned in a table instead of a text (lines 111-119)
Response: Thank you. A table with mentioned attributes and pharmacological actions found in literature based on most occurrence intensity, the antioxidant activity (AOA) at 30 min, and total amino acid (TAA) obtained in our study was presented.
- Section 3.3. Compounds, active principles, and industry importance of the local plants extracts literature synthesis contains a lot of redundant info; yes, the literature highlights those compounds in the tested extracts, however, the effect of the geographical origin, ecotypes, and seasonal variation could genuinely affect the chemical composition of the tested samples. This mandates that the authors should annotate their phytoconstituents.
Response: We also faced the same dilemma when we collected the analyzed materials for the synthesis. Although there is redundancy in the presentation of Compounds, active principles, and industry importance every time, there are significant differences not only from one species to another, but also different results can be observed depending on the geographical ecotypes. In addition, the compounds differ in intensity and appearance, so to avoid confusion, we have presented the compounds associated with antioxidant activity and possibly specific properties in the food industry each time.
Round 2
Reviewer 2 Report
The authors improved the manuscript